# Is ChatGPT a Good Multi-Party Conversation Solver?

**Chao-Hong Tan, Jia-Chen Gu, Zhen-Hua Ling**[*]

National Engineering Research Center of Speech and Language Information Processing,
University of Science and Technology of China, Hefei, China
chtan@mail.ustc.edu.cn, {gujc,zhling}@ustc.edu.cn

## Abstract

Large Language Models (LLMs) have emerged as influential instruments within the realm of natural language processing; nevertheless, their capacity to handle multi-party conversations (MPCs) – a scenario marked by the presence of multiple interlocutors involved in intricate information exchanges – remains uncharted. In this paper, we delve into the potential of generative LLMs such as ChatGPT and GPT-4 within the context of MPCs. An empirical analysis is conducted to assess the zero-shot learning capabilities of ChatGPT and GPT-4 by subjecting them to evaluation across three MPC datasets that encompass five representative tasks. The findings reveal that ChatGPT's performance on a number of evaluated MPC tasks leaves much to be desired, whilst GPT-4's results portend a promising future. Additionally, we endeavor to bolster performance through the incorporation of MPC structures, encompassing both speaker and addressee architecture. This study provides an exhaustive evaluation and analysis of applying generative LLMs to MPCs, casting a light upon the conception and creation of increasingly effective and robust MPC agents. Concurrently, this work underscores the challenges implicit in the utilization of LLMs for MPCs, such as deciphering graphical information flows and generating stylistically consistent responses.

## 1 Introduction

Large Language Models (LLMs), including notable instances such as ChatGPT (OpenAI, 2022) and GPT-4 (OpenAI, 2023), are ushering in a new era in the field of natural language processing (NLP), showcasing remarkable zero-shot and few-shot generalization capabilities. The methods of pretraining language models on extensive text corpora (Brown et al., 2020; Touvron et al., 2023), followed by alignment fine-tuning to ensure adherence to human instructions (Wang et al., 2023b;

Peng et al., 2023), has significantly amplified their proficiency in language comprehension, generation, interaction, and reasoning. The impressively high performance exhibited by these models has been extensively documented in related works (Qin et al., 2023; Wang et al., 2023a; Bubeck et al., 2023).

Multi-Party Conversations (MPCs) represent a prevalent and natural facet of human communication. In these exchanges, more than two participants engage in interactive discourse on a variety of topics. Such a dynamic introduces new challenges and opportunities for dialogue systems. Here, the key requirement isn't merely generating coherent and relevant utterances, but also making strategic determinations about when to intervene and whom to address in the conversation. We queried ChatGPT about its potential strategies for addressing the inherent challenges of MPCs with "*Can you solve multi-party conversation tasks?*". ChatGPT's response was as follows: "*I do not have built-in mechanisms to keep track of individual participants in a conversation. Therefore, it's important to explicitly mention the name or identifier of the participant you are addressing when providing instructions or asking questions.*", which was intriguing and touched upon some of the critical aspects we are investigating in this paper.

Considerable efforts have been made to explore the capabilities of LLMs across a variety of NLP tasks (Qin et al., 2023; Sun et al., 2023). Despite these advancements, the effectiveness of LLMs in handling MPCs remains largely underexplored. In this study, we scrutinize the potential of LLMs such as ChatGPT and GPT-4 in managing MPCs by implementing five distinct tasks including *Emotion Detection*, *Addressee Recognition*, *Speaker Identification*, *Response Selection*, and *Response Generation*, across three different MPC datasets. Our experiments reveal that both ChatGPT and GPT-4 can achieve performance on par with supervised methods when evaluated on the EmoryNLP

---

[*]Corresponding author.

and MELD datasets. However, the performance of these models on the more complex Ubuntu IRC dataset remains less than satisfactory. To address this, we introduce a strategy known as MPC Structure Incorporation, which weaves the speaker and addressee structure information of MPCs into the LLMs. This addition leads to a significant performance boost of ChatGPT and GPT-4 across all three datasets, underpinning the value of this approach for improving LLM effectiveness in MPC scenarios.

In summary, our contributions in this paper are three-fold: 1) An exploratory study to examine the performance of ChatGPT and GPT-4 in handling MPCs within a zero-shot context is carried out. This is the first study of its kind to investigate how these LLMs perform in MPC scenarios. 2) An MPC structure incorporation approach is proposed, which enhances the performance of ChatGPT and GPT-4 in managing MPCs. This strategy integrates the speaker and addressee structure information of MPCs into LLMs, leading to substantial performance improvements. 3) We delve into the potential of LLMs in handling MPCs and shed light on the challenges that need to be tackled in future research. This discussion forms the basis for continued investigations into improving LLM effectiveness in MPC contexts.

## 2 Related Work

**Large Language Models**   Recently, the development of large language models (LLMs) has made tremendous progress, as evidenced by models such as GPT-3 (Brown et al., 2020), PaLM (Chowdhery et al., 2022), LLaMA (Touvron et al., 2023), Chat-GPT (OpenAI, 2022), and GPT-4 (OpenAI, 2023). These LLMs exhibit emergent abilities, including in-context learning, mathematical reasoning, and commonsense reasoning (Wei et al., 2022). A recent line of work focuses on instruction learning, either via generating high-quality instructions data (Wang et al., 2023b; Peng et al., 2023) or by boosting LLMs with instructions (Chung et al., 2022; Taori et al., 2023).

**Multi-Party Conversations**   Existing methods on building MPC systems can be generally categorized into retrieval-based approaches (Ouchi and Tsuboi, 2016; Zhang et al., 2018; Wang et al., 2020; Gu et al., 2021, 2023) or generation-based (Hu et al., 2019; Gu et al., 2022; Li and Zhao, 2023). On the one hand, Ouchi and Tsuboi (2016)

and Zhang et al. (2018) proposed to update speaker embeddings with conversation streams dynamically and role-sensitively. Wang et al. (2020) proposed to track the dynamic topic in a conversation. Gu et al. (2021) proposed jointly learning "who says what to whom" in a unified framework by designing self-supervised tasks. Gu et al. (2023) present graph-induced fine-tuning which adapts Transformer-based LMs by integrating four types of edges into attention mechanisms. On the other hand, Hu et al. (2019) explored generation-based approaches by proposing a graph-structured network (GSN) that encoded the conversation context using homogeneous GNNs. Gu et al. (2022) proposed HeterMPC to model the complicated interactions between utterances and interlocutors with a heterogeneous graph. Li and Zhao (2023) proposed to iteratively generate missing addressees and optimize the generative model via the EM algorithm.

**LLMs for NLP Tasks**   Several contemporaneous papers present empirical studies of leveraging LLMs for various NLP tasks to explore whether LLMs can achieve competitive performance. For example, Qin et al. (2023) study the zero-shot learning capability of ChatGPT by evaluating it on 7 representative task categories, such as reasoning, natural language inference, and summarization. Liu et al. (2023) and Wang et al. (2023a) explore whether ChatGPT is a good evaluator of natural language generation. Sun et al. (2023) investigate LLMs for relevance ranking in information retrieval. Zheng et al. (2023) seek to understand why ChatGPT falls short in providing truthful answers and provide a guideline towards truthfulness in question answering via LLMs.

To the best of our knowledge, this paper makes the first attempt to empirically analyze the zero-shot learning ability of LLMs on the MPC tasks, providing comprehensive evaluation and analysis to inspire the development of more effective and robust MPC agents.

## 3 Approach

To explore an out-of-box MPC solver, our emphasis lies in the zero-shot setting. Instruction for each task is shown in Figure 1. For each task, LLMs are first instructed with the prompt "*You have been presented with a sequence of multi-party conversational turns, organized in chronological order.*". Then, LLMs are instructed to complete the task with task-specific prompts. To stabilize

Figure 1: The prompt template to complete the tasks.

Instruction:
*You have been presented with a sequence of multi-party conversational turns, organized in chronological order.*
${Task Instruction}
${Output Template}
${Extra Description}
*Use temperature=0, minimize unnecessary words to not get confused.*

Input:
${History with Input Template}
${Extend Instruction}

Response:

the output of LLMs, the effect of temperature is amplified with the prompt "*Use temperature=0, minimize unnecessary words to not get confused.*"[1] For more comprehensive explication, certain tasks require an *Extended Instruction* as shown in Table 5.

### 3.1 Task-Specific Prompts

Different instructions for each task are designed to guide LLMs to complete the task, namely task-specific prompts.

**Emotion Detection (ED)** LLMs are tasked to predict the emotion of each utterance with the task instruction "*Please evaluate the emotions of each utterance in the dialogue using the following $n$ labels: {...}.*" and the output template "*The output format must be: #{num} – {utterance} // {emotion}*" Here, $n$ is the number of emotion labels, and *{...}* is the list of emotion labels. Dialogue history is formalized as "*#{num} – {utterance}*".

**Addressee Recognition (AR)** LLMs are tasked to predict the addressee of each utterance with the task instruction "*Your task is to find the addressee of each utterance.*" and the output template "*The output format must be: #{num} – {utterance} // Reply to #{reply_i}*". And we can get the addressee information if we know the reply-to utterance of each utterance. Since the addressee of the first utterance is unknown, we add the extra description "*Please start from #1 since #0 is the first utterance that has no reply-to utterance. You should not leave any utterance unattended.*" to inform LLMs of the addressee of the first utterance. Dialogue history is formalized as the same as emotion detection.

**Speaker Identification (SI)** LLMs are tasked to predict the speaker of the last utterance with the

---

[1]We found that this prompt can improve the performance, even if the temperature is already set to 0 in API.

task instruction "*Please identify the speaker of the last sentence.*" and output template "*The output format should be only one speaker.*". Dialogue history with speaker is formalized as "*#{num} – {speaker}: {utterance}*".

**Response Selection (RS)** LLMs are tasked to select the most appropriate response from the candidates with the task instruction "*Your task is to select the most appropriate response from the candidate set.*" and output template "*The output format must be: #{num} – {utterance}*". The candidate set is formalized as "*#{num} – {utterance}*". Thus the history with input template is formalized as "*Dialogue History: {conversation turns} Candidates: {candidates}*".

**Response Generation (RG)** LLMs are tasked to generate a response with the task instruction "*Your task is to generate the most appropriate response.*". There is no need to provide the output template since the generation task is free-form.

### 3.2 MPC Structure Incorporation

Considering that the complicated interactions between interlocutors, between utterances and between an interlocutor and an utterance naturally increase the difficulty of fully understanding MPCs, it might be helpful to incorporate the MPC structure information.

**Speaker Structure** The speaker structure is incorporated into LLMs to help understand utterances. Specifically, the prompts with "*{utterance}*" is replaced with "*{speaker}: {utterance}*" to inform LLMs of the speaker of each utterance. An example is shown in Figure 2.

**Addressee Structure** The addressee structure of MPCs is constructed with adding sentence "*Reply to #{reply_uid} – {reply_utterance}*" into prompt.

**Speaker-Addressee Structure** The speaker-addressee structure of MPCs denotes the information flow from a speaker to an addressee, which is constructed with the prompt "*Reply to #{reply_uid} – Speaker {reply_spk}: {reply_utterance}*" and "*{utterance}*" is replaced with "*{speaker}: {utterance}*".

## 4 Experiments

### 4.1 Datasets

**EmoryNLP** EmoryNLP (Zahiri and Choi, 2018) is a collection of the TV show *Friends*, where

**Instruction:**
*You have been presented with a sequence of multi-party conversational turns, organized in chronological order. Please evaluate the emotions of each utterances in the dialogue using the following 7 labels: {'fear', 'neutral', 'joy', 'disgust', 'anger', 'sadness', 'surprise'}. The output format must be: #{num} -- {speaker}: {utterance} // {emotion} Use temperature=0, minimize unnecessary words to not get confused.*

**Input:**
#0 -- All: Push, push!
#1 -- Ross: We're here!
#2 -- Carol: Where have you been?
#3 -- Ross: Long story, honey.
#4 -- Dr. Franzblau: All right, Carol, I need you to keep pushing. I need
— Excuse me, could I have this?

**Response:**
#0 -- All: Push, push! // neutral
#1 -- Ross: We're here! // joy
#2 -- Carol: Where have you been? // neutral
#3 -- Ross: Long story, honey. // neutral
#4 -- Dr. Franzblau: All right, Carol, I need you to keep pushing. I need
— Excuse me, could I have this? // neutral

Figure 2: The prompt and output of emotion detection with speaker structure.

each utterance is annotated with one of the seven emotions borrowed from the six primary emotions in the Willcox (1982)'s feeling wheel, including *sad, mad, scared, powerful, peaceful, joyful*, and a default emotion of *neutral*.

**MELD** The Multimodal EmotionLines Dataset (MELD) (Poria et al., 2019) was developed by improving and expanding the original EmotionLines dataset (Hsu et al., 2018). MELD includes the same multi-party dialogue instances as EmotionLines but incorporates additional audio and visual modalities alongside text. Each utterance within a dialogue is tagged with one of the seven emotions, including: *Anger, Disgust, Sadness, Joy, Neutral, Surprise, Fear*. This comprehensive tagging facilitates deep emotion-focused analysis.

**Ubuntu IRC benchmark** Ubuntu IRC (Hu et al., 2019) represents a substantial, Ubuntu Internet Relay Chat (IRC) channel corpus, replete with annotations for multiple interlocutors. Furthermore, it is distinguished by the provision of addressee labels accompanying each individual utterance.

The EmoryNLP and MELD datasets proffer emotion labels, hence enabling the execution of ED tasks. Conversely, the Ubuntu IRC dataset offers Addressee tags, thereby serving as a resource for AR tasks. Table 1 presents the statistics of the three datasets evaluated in our experiments.

| Datasets | Train | Valid | Test |
|---|---|---|---|
| EmoryNLP (Zahiri and Choi, 2018) | 659 | 89 | 79 |
| MELD Poria et al. (2019) | 1,039 | 114 | 280 |
| Ubuntu IRC (Hu et al., 2019) | 311,725 | 5,000 | 5,000 |

Table 1: Statistics of the datasets evaluated in this paper.

### 4.2 Baselines

**(1) BERT** (Devlin et al., 2019) is a bidirectional language representation model and can be fine-tuned for various NLP tasks. **(2) GPT-2** (Radford et al., 2019) is a uni-directional pre-trained language model. **(3) BART** (Lewis et al., 2020) is a denoising autoencoder using a standard Tranformer-based architecture, trained by corrupting text with an arbitrary noising function and learning to reconstruct the original text. **(4) SPCL-CL-ERC** (Song et al., 2022) introduced a novel Supervised Prototypical Contrastive Learning loss function for the Emotion Recognition in Conversation task, addressing the issues stemming from imbalanced classification through the medium of contrastive learning, obviating the necessity for large batch sizes. It is the SOTA of the EmoryNLP and MELD dataset on ED task. **(5) BART w/. EM** (Li and Zhao, 2023) introduce an Expectation-Maximization (EM) method that alternately performs the expectation steps to infer addressee labels, and the maximization steps to fine-tune a response generation model. It is the SOTA of the Ubuntu IRC datasets on RG task. **(6) MPC-BERT w/. GIFT** (Gu et al., 2021, 2023) is a pre-trained model for MPCs that learns 'who says what to whom' with self-supervised tasks (MPC-BERT) and a graph-based fine-tuning method (GIFT). It is the SOTA of the Ubuntu IRC datasets on AR, SI and RS tasks. **(7) ChatGPT** (OpenAI, 2022), ingeniously enriched by the infusion of Supervised Fine-Tuning (SFT) and Reinforcement Learning from Human Feedback (RLHF) methodologies, ensuring seamless synchronization between the model and human directives. **(8) GPT-4** (OpenAI, 2023) is a large-scale, multimodal model which can accept image and text inputs and produce text outputs, exhibiting human-level performance on various professional and academic benchmarks. [2]

### 4.3 Implementation Details

All supervised models were trained with the AdamW method (Loshchilov and Hutter, 2019).

---

[2]*ChatGPT* and *GPT-4* are recognized representatives of LLMs, so we only consider them to evaluate MPCs.

The learning rate was initialized as $6.25e$-$5$ and was decayed linearly down to $0$. The batch size was set to $128$ gradient accumulation steps. Models were trained in 10 epochs. For ChatGPT and GPT-4, we used the API endpoints *gpt-3.5-turbo-0301* and *gpt-4-0314* provided by OpenAI respectively.[3]

## 4.4 Metrics

To evaluate ED task, we employed the weight-F1 score, which is the harmonic mean of precision and recall. To evaluate SI and AR tasks, we employed accuracy. To evaluate RS task, we employed R10@1, which is the percentage of the first correct response selected from 10 candidates. To evaluate the quality of the generated text, we employed the standard string-similarity-based metrics SacreBLEU (Post, 2018), ROUGE (Lin, 2004) and METEOR (Banerjee and Lavie, 2005). Higher is better for all metrics. All metrics were calculated by the `evaluate` toolkit [4].

## 4.5 Evaluation Results of MPC Understanding

As shown in Table 2, we evaluated the dialogue understanding performance of supervised language models and LLMs on three test sets. Here, the SOTA results of EmoryNLP and MELD on ED are copied from Song et al. (2022). The SOTA results of Ubuntu IRC of all three tasks are copied from Gu et al. (2023).

**Supervised Language Models Versus LLMs** When one juxtaposes the outcomes associated with supervised language models and LLMs, it becomes apparent that the LLMs demonstrate parity in performance with their supervised counterparts on the EmoryNLP and MELD datasets. However, their performance on Ubuntu IRC falls short of the mark. It is unsurprising to discern that the capability of *GPT-4* surpasses its predecessor, *ChatGPT* across all four understanding tasks. In the ED task, both *ChatGPT* and *GPT-4* outperform BERT but fall short of the state-of-the-art (SOTA) on EmoryNLP. Additionally, *ChatGPT* lags behind BERT on the MELD dataset. For the AR task, both *ChatGPT* and *GPT-4* trail behind BERT and SOTA on the Ubuntu IRC dataset, respectively. Regarding the SI

task, it is essential to note that speaker information detection is unattainable without the provision of explicit speaker information. Consequently, our evaluation of *ChatGPT* and *GPT-4* in the SI task is limited to instances where speaker information is provided. Results demonstrate that both models outperform BERT on both EmoryNLP and MELD datasets. Specifically, *ChatGPT* and *GPT-4* exhibit superior performance to BERT by 3.60% and 16.46% on EmoryNLP, and by 13.49% and 34.42% on MELD, respectively. However, it is worth noting that *ChatGPT* and *GPT-4* significantly trail behind supervised models on the Ubuntu IRC dataset. Regarding the RS task, only in the context of the MELD dataset does *GPT-4* outshine BERT. Specifically, on the SI task of the Ubuntu IRC dataset, *ChatGPT* and *GPT-4* lag behind BERT by 25.58% and 12.31%, respectively. This can be attributed to the fact that Ubuntu IRC leans towards a more technical and specialized domain, which is also difficult for humans to understand.

**Speaker Information Enhancement** To delve into the significance of the interlocutor within the MPC understanding, we incorporate speaker information into three distinctive tasks across three datasets, disregarding the ED task. Comparing the line of *ChatGPT (GPT-4)*[5] and *ChatGPT (GPT-4) w/. Speaker*, we can find that the incorporation of speaker information can improve the performance of *ChatGPT (GPT-4)* on all five tasks except for the RS task of Ubuntu IRC. This improvement is not surprising. In the context of MPCs, the speaker doesn't adhere to the rigid alternation characteristic of two-party dialogues, hence the incorporation of speaker information can enhance the lucidity of the conversation, rendering it more readily comprehensible. Note that the substantial advancement observed in the AR task partly stems from our modification of the task from identifying the response sentences to directly recognizing the addressee. Nonetheless, compared with *ChatGPT*, the subpar performance of *ChatGPT w/. Speaker* on the RS task of Ubuntu IRC suggests that the imparted speaker information could not be optimally assimilated and deployed by *ChatGPT*. On the contrary, it appeared to disrupt the process of response selection. However, *GPT-4*'s performance on this task was substantially superior, enhancing the effectiveness of the RS task by a margin of

---

[3]Code is available at https://github.com/lxcht an/ChatMPC.

[4]https://github.com/huggingface/evalu ate

[5]This syntax denotes that the conclusion is applicable to both *ChatGPT* and *GPT-4*.

| Tasks \ Models | EmoryNLP | | | MELD | | | Ubuntu IRC | | |
|---|---|---|---|---|---|---|---|---|---|
| | ED (F1) | SI (ACC) | RS (R10@1) | ED (F1) | SI (ACC) | RS (R10@1) | AR (ACC) | SI (ACC) | RS (R10@1) |
| **Supervised** | | | | | | | | | |
| BERT (Devlin et al., 2019) | 34.76 | 47.82 | 48.68 | 61.31 | 44.18 | 48.13 | 82.88 | 71.81 | 73.42 |
| SOTA | 40.94 | – | – | **67.25** | – | – | **90.18** | **90.50** | **80.74** |
| **LLMs** | | | | | | | | | |
| ChatGPT (OpenAI, 2022) | 37.16 | – | 29.11 | 58.32 | – | 36.42 | 54.11 | – | 54.68 |
| w/. Speaker | 38.50 | 51.42 | 34.21 | 60.90 | 57.67 | 39.17 | 67.19 | 46.23 | 49.52 |
| w/. Addressee | – | – | – | – | – | – | – | – | 37.50 |
| w/. Speaker & Addressee | – | – | – | – | – | – | – | 38.50 | 29.50 |
| GPT-4 (OpenAI, 2023) | 39.38 | – | 44.30 | 62.32 | – | 53.73 | 66.02 | – | 61.50 |
| w/. Speaker | **41.63** | **64.28** | **50.00** | 64.18 | **78.60** | **57.46** | 82.50 | 58.00 | 71.00 |
| w/. Addressee | – | – | – | – | – | – | – | – | 68.50 |
| w/. Speaker & Addressee | – | – | – | – | – | – | – | 59.50 | 72.00 |

Table 2: Evaluation results of MPC Understanding. Numbers in **bold** denoted that the results achieved the best performance. The vacant cells signify their incalculability.[6] The SOTA of ED task is *SPCL-CL-ERC* (Song et al., 2022), and the SOTA of Ubuntu IRC tasks is *MPC-BERT w/. GIFT* (Gu et al., 2023)

| Metrics (RG) \ Models | EmoryNLP | | | MELD | | | Ubuntu IRC | | |
|---|---|---|---|---|---|---|---|---|---|
| | S-BLEU | ROUGE$_L$ | METEOR | S-BLEU | ROUGE$_L$ | METEOR | S-BLEU | ROUGE$_L$ | METEOR |
| **Supervised** | | | | | | | | | |
| GPT-2 (Radford et al., 2019) | 0.6175 | 7.90 | 10.26 | 0.5160 | 6.01 | 7.74 | 0.93 | 9.53 | 4.01 |
| BART (Lewis et al., 2020) | 0.7009 | 8.63 | 11.86 | **1.0757** | 8.64 | 10.37 | 0.95 | 9.90 | 4.46 |
| SOTA | – | – | – | – | – | – | **2.45** | **11.71** | 5.52 |
| **LLMs** | | | | | | | | | |
| ChatGPT (OpenAI, 2022) | 0.5358 | 9.03 | 11.30 | 0.9059 | 7.13 | 8.63 | 0.5145 | 8.64 | 8.85 |
| w/. Speaker | 0.3082 | 8.95 | 11.45 | 0.9159 | 8.17 | 9.86 | 0.8512 | 9.68 | 9.45 |
| w/. Addressee | – | – | – | – | – | – | 0.3332 | 10.73 | 11.49 |
| w/. Speaker & Addressee | – | – | – | – | – | – | 0.6101 | 10.60 | 12.07 |
| GPT-4 (OpenAI, 2023) | 0.4608 | 8.60 | 12.22 | 0.9301 | 8.93 | 12.43 | 0.2206 | 7.86 | 12.03 |
| w/. Speaker | **0.9049** | **9.99** | **14.61** | 0.9666 | **8.89** | **13.06** | 0.2324 | 9.18 | 12.65 |
| w/. Addressee | – | – | – | – | – | – | 0.2458 | 9.52 | 12.78 |
| w/. Speaker & Addressee | – | – | – | – | – | – | 0.2856 | 9.24 | **13.30** |

Table 3: Evaluation results of MPC Generation. Numbers in **bold** denoted that the results achieved the best performance. S-BLEU is the short of SacreBLEU. The vacant cells signify their incalculability.[6] The SOTA of Ubuntu IRC is *BART w/. EM* (Li and Zhao, 2023).

9.50%. This result suggests that *GPT-4* is better at fusing speaker information than *ChatGPT*.

**Addressee Information Enhancement**   To probe the import of the addressee within the context of MPC comprehension, we integrate addressee information into the SI and RS tasks of Ubuntu IRC. This is primarily due to the absence of addressee information within the other two datasets. The AR task is excluded from this process, given that addressee information is unavailable within the corresponding Ubuntu IRC task. When drawing comparisons between the results of *ChatGPT w/. Speaker* and *ChatGPT w/. Speaker & Addressee* on

the SI and RS tasks, as well as between the results of *ChatGPT* and *ChatGPT w/. Addressee* on the RS task, we find, to our surprise, that the integration of addressee information has led to a diminution in performance. When comparing the results of *GPT-4 w/. Speaker* and *GPT-4 w/. Speaker & Addressee* on SI and RS tasks, we find that the incorporation of addressee information can slightly improve the performance. The only marginal improvement of approximately 1.00% observed on the SI and RS tasks of Ubuntu IRC can be predominantly attributed to the proficiency of *GPT-4 w/. Speaker* in correctly inferring addressee information, given its noteworthy accuracy of 82.50% on AR tasks.

**Speaker and Addressee Information Enhancement**   When contrasting the results of *ChatGPT (GPT-4) w/. Speaker & Addressee* with the *ChatGPT (GPT-4)* on RS tasks, we discern that the

---

[6]For SI task, the speakers in dialogue history are needed. It means that we cannot lack SI information, so these cells are empty. For EmoryNLP and MELD datasets lacking the addressee information, therefore corresponding rows are empty. For the Ubuntu IRC AR task, it asked not to add the addressee information, so these cells are empty.

---

### Instruction:
You have been presented with a sequence of multi-party conversational turns, organized in chronological order.
Please identify the speaker of the last sentence. The output format should be only one speaker.
Use temperature=0, minimize unnecessary words to not get confused.
Note that the speaker is one of ['Speaker 1', 'Speaker 2', 'Speaker 3', 'Speaker 4'].

### Input:
#0 – Speaker 1: let me look at this post-meeting
[Reply to #0 ... ] #1 – Speaker 2: are you doing promotions to main ? if so when is a good time to ping you about the xfce lot ?
[Reply to #1 ... ] #2 – Speaker 1: after we have anastacia again so that i do n't break stuff
...
[Reply to #5 ... ] #6 – have FILEPATH checked for reverse-dependencies ? the archive tools to do that check do n't work yet

### Response:

Answer: Speaker 1
*GPT-4 w/. Speaker & Addressee*: Speaker 1
*ChatGPT w/. Speaker & Addressee*: Speaker 6
*GPT-4 w/. Speaker*: Speaker 4
*ChatGPT w/. Speaker*: Speaker 4

---

Table 4: One selected example of the SI task on Ubuntu IRC with speaker and addressee structure. Here we also list the results of *ChatGPT (GPT-4) w/. Speaker* for comparison.

integration of speaker and addressee information precipitates a performance decrement of 25.18% for *ChatGPT*, whilst conversely, it elicits an augmentation of 11.50% in the performance of *GPT-4*. Indeed, both categories of information serve to enhance the comprehension of the conversation, yet the infusion of additional data concurrently amplifies the complexity of understanding. Empirical outcomes indicate that *ChatGPT* grapples with the processing of this surplus information, whereas its more potent successor, *GPT-4*, exhibits the capacity to assimilate it effectively.

### 4.6 Evaluation Results of MPC Generation

As shown in Table 3, we evaluated the dialogue response generation performance of supervised language models and LLMs on three test sets. The SOTA results of Ubuntu IRC of all three tasks are copied from Li and Zhao (2023).

**Supervised Language Models Versus LLMs** It becomes apparent that the SacreBLEU scores of supervised models consistently eclipse those of LLMs in a significant number of cases across all three evaluative subsets. This phenomenon is largely a byproduct of ChatGPT and GPT-4's inclination to generate more prolix responses, an approach inherently detrimental to the calculation of SacreBLEU. Regarding the $ROUGE_L$ metrics, LLMs yield superior results compared to supervised models on the EmoryNLP and MELD test sets. In contrast, within the ambit of the Ubuntu IRC dataset, supervised models command a greater presence—an outcome likely driven by the amplified comprehension complexity associated with

this particular set. In terms of the METEOR metric, LLMs outdistance supervised models across all test sets, thereby affirming the formidable aptitude of LLMs in generating responses.

**MPC Structure Incorporation** Similar to the results in MPC Understanding, the inclusion of interlocutors' information emerges as beneficial to the generation of dialogue responses. This is substantiated by the marked superiority of *ChatGPT (GPT-4) w/. Speaker* over *ChatGPT (GPT-4)* across all three test sets, with the exception of *ChatGPT*'s performance on the EmoryNLP dataset. Nonetheless, addressee information doesn't seem to confer a discernible advantage in the generation of dialogue responses when comparing the performance of *ChatGPT w/. Addressee* to the *ChatGPT*, as well as the performance of *ChatGPT w/. Speaker & Addressee* to *ChatGPT w/. Speaker*. When our focus shifts to *GPT-4*, we observe that the performance of *GPT-4 w/. Addressee* does not exhibit the anticipated enhancement. However, with the addition of interlocutor information, there is a noticeable improvement in the performance of *GPT-4 w/. Speaker & Addressee*.

### 4.7 Case Study

To analyze the performance of LLMs on the tasks of MPC understanding and generation specifically, case studies were conducted by presenting randomly selected examples for further illustration.

**MPC Understanding** As shown in Table 4, in the absence of addressee information, the model exhibits inadequacies in comprehending MPC,

### Instruction:
You have been presented with a sequence of multi-party conversational turns, organized in chronological order.
Your task is to generate the most appropriate response.
The output format is "[Reply to #{reply_uid} – Speaker {reply_spk}: {reply_utterance}] Speaker {rsp_spk}: {rsp}".
Use temperature=0, minimize unnecessary words to not get confused.

### Input:
–
Dialogue History:
#0 – Speaker 1: yep - only 4 primaries.. but ive never needed more then 4
[Reply to #0 ... ] #1 – Speaker 2: but you ca n't see your other partitions right ?
[Reply to #3 ... ] #4 – Speaker 2: i see there 's some confusion , that 's my mistake i guess.. with active i meant that
windows ca n't 'see ' the other partitions.. but i can see that 's only a historical issue..
–
Please give a response on behalf of Speaker 1 for Uttenrance #4.
The part of response is [Reply to #4 – Speaker 2: i see there 's some confusion , that 's my mistake i guess.. with active i
meant that windows ca n't 'see ' the other partitions.. but i can see that 's only a historical issue..] Speaker 1.
Please finish the response generation.

### Response:

| BART | i'm not sure, but i'm sure there is a way to do that |
|---|---|
| ChatGPT | You can automate the postfix setup routine and use one machine as the relay for all the others. This will save you time and effort. |
| w/. Speaker & Addressee | Yes, you can automate the setup process by creating a script that answers the questions for you. This will save you time and ensure that the setup is consistent across all machines. |
| GPT4 | If you're trying to automate the process and distribute it to multiple machines, you might consider creating a shell script or Ansible playbook that sets up everything necessary for node.js development, including postfix. This way, you can handle the postfix setup with pre-defined configurations, and then package all the necessary files and tools into an AMI for EC2 or an Ubuntu image. This would be more maintainable and scalable. |
| w/. Speaker & Addressee | Yes, you can automate the process. You can use tools like 'expect' or 'autoexpect' in Linux to handle interactive command line sessions. These tools can learn from your inputs and can generate a script that automates the process. Alternatively, you might also consider using a configuration management tool like Ansible or Puppet. |
| Ground Truth | and untar your default config on all your client machines . |

Table 5: One selected example of the RG task on Ubuntu IRC with speaker and addressee structure. Here we also list the results of ChatGPT, GPT-4 and BART for comparison.

resulting in inaccurate responses from both *Chat-GPT* and *GPT-4*. However, when furnished with addressee information, *GPT-4* shows an improved aptitude to comprehend the dialogue, leading to a correct response. Conversely, *ChatGPT* appears to be confounded by the addressee information, yielding an incorrect answer that lies beyond the purview of the candidates.

**MPC Generation** As shown in Table 5, the response generated by BART is conspicuously devoid of substantive content. All the LLMs, especially *GPT-4*, exhibit the ability to produce lengthy and meaningful responses. Although *ChatGPT* and *GPT-4* are capable of crafting responses with greater pertinence, their propensity to yield verbose responses hampers the SacreBLEU score.

## 5 Conclusion

In this paper, we have explored the abilities of generative LLMs for MPCs, which have been largely underexplored. We empirically analyze the zero-shot learning ability of ChatGPT and GPT-4 by evaluating them on three popular MPC datasets covering five representative tasks. On EmoryNLP and MELD datasets, ChatGPT and GPT4 achieve comparable performance to the supervised training models. However, ChatGPT performs poorly on the evaluated Ubuntu IRC tasks, while GPT-4 shows promising results. However, there is still a large gap relative to the supervised training on SI task of Ubuntu IRC. Taking into account the structure of the MPC, it is evident that both ChatGPT and GPT-4 exhibit enhanced performance across nearly all tasks when equipped with speaker information. However, in the context of addressee information, ChatGPT's performance may decline if it is encumbered with extraneous data. Conversely, GPT-4 aptly leverages this information to accomplish the task with heightened proficiency. Devoting efforts towards efficaciously intertwining the graphical structure inherent in MPCs with LLMs through prompts or even supervised fine-tuning constitutes one of future work.

## Limitations

The design of prompts plays a crucial role in determining the final results, as it holds significant sway over the outcome. The way in which prompts are constructed and structured can greatly impact the performance and effectiveness of ChatGPT and GPT-4 when it comes to handling MPC tasks. However, our current prompt architecture might not fully encompass the ideal potential and capabilities of these advanced language models in tackling MPC tasks. There is a possibility that further improvements and refinements in the prompt design can unlock even greater performance and unleash the full potential of ChatGPT and GPT-4 in handling MPC tasks. It is essential to explore and enhance the prompt architecture to ensure optimal results and leverage the capabilities of these powerful language models to their fullest extent in the realm of MPC tasks.

## Acknowledgements

This work was supported by the Opening Foundation of State Key Laboratory of Cognitive Intelligence, iFLYTEK COGOS-2022005. We thank anonymous reviewers for their valuable comments.

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
