# OpenReview forum: "Is ChatGPT a Good Multi-Party Conversation Solver?"
_EMNLP/2023/Conference — EMNLP 2023 Findings_

### Official Review · Reviewer_2d2w · 2023-08-04

**Soundness:** 3

**Excitement:**

3: Ambivalent: It has merits (e.g., it reports state-of-the-art results, the idea is nice), but there are key weaknesses (e.g., it describes incremental work), and it can significantly benefit from another round of revision. However, I won't object to accepting it if my co-reviewers champion it.

**Paper Topic And Main Contributions:**

This paper investigates how well zero-shot methods using ChatGPT and GPT-4 can perform on multi-party conversation (MPC) tasks compared to supervised models.
The authors design prompts for zero-shot learning and evaluate tasks such as dialogue understanding and response generation. The authors also examine the effects of including speaker and addressee information in the prompt.
The corpora used are EmoryNLP, MELD, and Ubuntu IRC, which were not newly constructed for this research.
The contribution of this paper lies in examining the dialogue capabilities of recently introduced LLMs in situations such as MPC, which are thought to be more challenging than regular one-on-one conversations.

**Reasons To Accept:**

- Since the use of LLMs in MPC is expected to expand in the near future, quantitatively evaluating their dialogue capabilities is important.
- The current research appropriately compares these capabilities with existing supervised methods.

**Reasons To Reject:**

- Although the main focus of this paper is on evaluation, it does not offer much in the way of technical novelty. It just designs the prompt given to the LLM.
- Evaluations are conducted using closed models like ChatGPT and GPT-4, so even if the temperature is lowered, there's a possibility that the models themselves could be updated frequently, which means that the consistency of results cannot be guaranteed.

**Reproducibility:**

3: Could reproduce the results with some difficulty. The settings of parameters are underspecified or subjectively determined; the training/evaluation data are not widely available.

**Reviewer Confidence:**

4: Quite sure. I tried to check the important points carefully. It's unlikely, though conceivable, that I missed something that should affect my ratings.

**Typos Grammar Style And Presentation Improvements:**

Section 4.1 "iis" -> "is"
Table 3 "Evaluation results of MPC Understanding." -> "Evaluation results of MPC Generation."

---

> ### Author Rebuttal · Authors · 2023-08-29
>
> We appreciate your feedback and insights.
>
> #### **Re** *Reason To Reject 1*:
>
> *"Although the main focus of this paper is on evaluation, it does not offer much in the way of technical novelty. It just designs the prompt given to the LLM."*
>
> 1. Since ChatGPT is a closed source model, we only have access to its inputs as well as outputs. So what we can do is change its inputs (Prompts) to get better results.
> 2. The main purpose of this paper is to explore whether the most powerful language models available can be good tools for solving MPC problems, which have not yet been explored. We believe that this paper can provide good inspiration for follow-up work.
> 3. MPCs do not maintain sequential information flows since the third utterance can reply to the first one, which brings a challenge for autoregressive serialized LLMs, especially for ChatGPT, which is carefully designed for two-person conversations.
> 4. As for how to improve the performance of LLMs in MPC, inspired by MPC--BERT [1], we utilize additional information including the speaker and the addressee. How these two types of information are utilized and how they affect the performance of LLMs in MPC tasks are also our contributions. In addition, we discuss the experimental results in detail. This discussion forms the basis for continued investigations into improving LLM effectiveness in MPC contexts.
>
> #### **Re** *Reason To Reject 2*:
>
> *"Evaluations are conducted using closed models like ChatGPT and GPT-4 ..."*
>
> 1. Since NLP research is constantly evolving, the increasing performance of LLMs is obvious. Our evaluation is for the strongest LLMs of the day.
>
> 2. We use *gpt-4-0314* and *gpt-3.5-turbo-0301*APIs provided by OpenAI. As stated in OpenAI docs, these models will not receive updates, and will be maintained to June 13th 2024 at the earliest, which means that we can get the consistency of results in about 1 year.
>
> #### **Re** *Typos*:
>
> Thank you and we will correct them in the revision.
>
> #### **References**
>
> [1] MPC-BERT: A Pre-Trained Language Model for Multi-Party Conversation Understanding. Gu et al. ACL 2021.

---

### Official Review · Reviewer_CWvj · 2023-08-04

**Soundness:** 2

**Excitement:**

2: Mediocre: This paper makes marginal contributions (vs non-contemporaneous work), so I would rather not see it in the conference.

**Paper Topic And Main Contributions:**

This paper investigates the performance of large language models on 3 multi-party dialog datasets for 5 tasks: emotion detection, addressee recognition, speaker identification, response selection, response generation. The models are put into a position of overhearing and queried for their ability to follow the conversation (speaker and addressee identification), judge contributions (emotion recognition), or take the position of one of the participants (response selection and generation).

**Questions For The Authors:**

A: In Figure 2 and in Section 3.1, you mention "extend instruction" and I do not understand what that means, especially since none of the 5 tasks seems to need this item. Do you maybe mean "extended instruction" as in "more details about the task"? How does this differ from "task instruction" and "extra description"?

B: I am having trouble reading Table 2, especially all the empty cells: For the base LLM models, there are no results reported for SI, why is that? Why are there no results for adding speaker and adding speaker and addressee information?

C: The prompt does not contain any explanation of the input format (e.g. it does not explain what the #1, #2 etc stands for or what the variables like rsp mean). Have you checked that the models can interpret the format correctly?

**Reasons To Accept:**

- The work tests an interesting ability of LMs – their ability to keep track of a multi-party conversation.

**Reasons To Reject:**

- The main reason to reject is that we do not know anything about the training data of the models that are tested here. It is unclear whether the evaluated datasets have been in the training data of ChatGPT or GPT-4 and therefore the results are unreliable and not generalizable.

- There is no in-depth error analysis of failure cases. It would be interesting to see whether errors are systematic and how they differ from the failure and success cases of the supervised models.

- The framing that the title suggests ("to solve multi-party conversation") is misleading. All tasks aim at being able to follow a conversation (speaker and addressee identification), judge contributions (emotion recognition), or take the position of one of the participants (response selection and generation). The latter two tasks give the impression of the LMs actively participating in the conversation but the evaluation aims at their ability to reproduce the ground truth. (i.e. a generated contribution could be acceptable in a dialog even if counted as wrong here).

**Update after rebuttal**

I thank the authors for their comments and additional explanations and for providing answers to my questions. I have updated my scores. While I appreciate the methodology and chosen evaluation tasks as well as the decision to leave more in-depth analysis for later, my main concern remains the first reason to reject above.

**Reproducibility:**

4: Could mostly reproduce the results, but there may be some variation because of sample variance or minor variations in their interpretation of the protocol or method.

**Reviewer Confidence:**

4: Quite sure. I tried to check the important points carefully. It's unlikely, though conceivable, that I missed something that should affect my ratings.

**Typos Grammar Style And Presentation Improvements:**

- Section 4 "Speaker Information Enhancement": I do not understand this syntax "GPT-3.5-Turbo (GPT-4)". There are several instances of these parentheses, it is unclear to me what this means.
- The actual tasks are not mentioned until Section 3 which made reading the related work section odd bcecause I didn't know what the target for the relation was. Please mention the tasks in the introduction (naming them is sufficient in the introduction).
- Table 3: Please spell out the abbreviated tasks in the caption for easier mapping. Also please mention what exactly SOTA refers to (i.e. what paper).
- Section 4.1, EmoryNLP: "iis" -> "is", MELD: "enrichadding to the richness"
- Table 2: It is confusing that the third column for Ubuntu doesn't match the other two (instead ED it reports AR), please highlight this in the caption or the header. Also please spell out what SOTA refers to, e.g. in the caption.
- I would find it easier if the settings had some kind of special syntax like using italics or subscript instead of regular text because the names get quite long and often span line breaks, e.g. in section "MPC Structure Incorporation" ("GPT-3.5-Turbo (GPT-4) w/. Speaker", "GPT-3.5-Turbo w/. Addressee", etc.)
- In the related work section you mention a number of LLMs but I did not understand why you picked only GPT-4 and ChatGPT to test.
- I found "addressee recognition" to be confusing: addressee in dialog can also be interpreted in the forward-looking sense. I was initially confused about the first utterance not having an addressee because surely the first utterance is addressed to someone. it is just not replying to anyone. this is better phrased differently

---

> ### Author Rebuttal · Authors · 2023-08-29
>
> We appreciate your feedback and insights.
>
> #### **Re** *Reason To Reject 1*:
>
> *"The main reason to reject is that we do not know anything about the training data of the models that are tested here. It is unclear whether the evaluated datasets have been in the training data of ChatGPT or GPT-4 and therefore the results are unreliable and not generalizable."*
>
> Your apprehensions are warranted, yet we proffer the following perspectives.
>
> 1. Most datasets have been accessed by LLMs in pretraining stage.
> 2. Most of the existing research works[1-2] on LLMs performance do not consider this problem.
> 3. Since LLMs has seen too much data, the probability of public datasets being accessed is large. However, even though the LLMs pre-training process has access to this data, there are no supervised labels to train the task on, and the task is considered unseen.
> 4. We test on multiple MPC datasets to reduce the bias introduced by a single dataset.
>
> #### **Re** *Reason To Reject 2*:
>
> *"There is no in-depth error analysis of failure cases. It would be interesting to see whether errors are systematic and how they differ from the failure and success cases of the supervised models."*
>
> 1. I appreciate your interest in a more in-depth error analysis of failure cases in our study.
>
> 2. While we value the feedback, I would like to provide a rebuttal by explaining our rationale for not including such an analysis in the current study. Our study was primarily focused on evaluating the performance of the LLMs in solving MPC problems. Due to the scope and focus of our study, we believe that such an analysis would be more suitable as a separate research endeavor. We hope that our study provides a valuable contribution to the field and encourages further research in this area, including more comprehensive error analyses in future investigations.
>
> #### **Re** *Reason To Reject 3*:
>
> *"The framing that the title suggests ("to solve multi-party conversation") is misleading. ..."*
>
> We adopt the above statement due to the following considerations.
>
> 1. The difficulty with MPC is that the stream of conversations has complex relationships, and determining who speaks (Speaker) and speaker to whom (Addressee) is key to recognizing the relationships.
> 2. In terms of tasks coverage, we study two major parts including MPC understand and MPC response generation. For the MPC understanding task, we refer to some papers[3-4] and select representative ED, AR, SI, RS as evaluation. Therefore, we believe that our evaluation is sufficient to cover most of the problems of MPC.
>
> *"The latter two tasks give the impression of the LMs actively ..."*
>
> We evaluated the performance of response generation using metrics that are common in response generation tasks, which are also used in papers on MPC.
>
> #### **Re** *Question 1:*
>
> *"In Figure 2 and in Section 3.1, you mention "extend instruction" and I do not understand what that means, especially since none of the 5 tasks seems to need this item. Do you maybe mean "extended instruction" as in "more details about the task"? How does this differ from "task instruction" and "extra description"?"*
>
> Sorry for the unclear presentations. "extend instruction" is used for incorporating additional information into MPCs. For example, as shown in Table 5, the paragraph “Please give a response on behalf of Speaker …” is "extend instruction".
>
> #### **Re** *Question 2:*
>
> *"I am having trouble reading Table 2, especially all the empty cells: For the base LLM models, there are no results reported for SI, why is that? Why are there no results for adding speaker and adding speaker and addressee information?"*
>
> Thank you for your suggestion, we will add detailed notes to Table 2 in our revision. Here's why empty cells exist.
>
> 1. For the speaker identification task, we need to determine which speaker has spoken the current sentence from the history of speakers and what they have said. This means that we cannot lack SI information, so these cells are empty.
> 2. For EmoryNLP and MELD datasets, they did not provide the Addressee information, therefore *w /. Addressee* and *w /. Speaker & Addressee* two rows are empty.
> 3. For the Ubuntu IRC AR task, we need to estimate the Addressee, so we can't add the Addressee information, resulting in the cells with Addressee information are empty.
>
> #### **Re** *Question 3:*
>
> *"The prompt does not contain any explanation of the input format (e.g. it does not explain what the #1, #2 etc stands for or what the variables like rsp mean). Have you checked that the models can interpret the format correctly?"*
>
> The models are able to understand these special symbols. For example, in the ED task of MELD dataset, we have an output "#0 -- Hi! // joy\n#1 -- Hi. // neutral". We will add more cases to the appendix of the revised version to show the raw output of the model.
>
> #### **Re** *Typos 1:*
>
> *"Section 4 "Speaker Information Enhancement": I do not understand this syntax "GPT-3.5-Turbo (GPT-4)". There are several instances of these parentheses, it is unclear to me what this means."*
>
> It means that both the GPT-3.5-Turbo and GPT-4 have the same conclusions.
>
> #### **Re** *Typos 2:*
>
> *"The actual tasks are not mentioned until Section 3 ..."*
>
> Thank you for your advice. We will add them in Section Introduction in the revision.
>
> #### **Re** *Typos 3:*
>
> *"Table 3: Please spell out the abbreviated tasks in the caption for easier mapping. Also please mention what exactly SOTA refers to (i.e. what paper)."*
>
> We explained all the SOTA in Section 4.2 Baselines. Thank you for your advice. Different tasks have different SOTA methods, and we will add instructions in Table 2 to make it clearer.
>
> #### **Re** *Typos 4:*
>
> *"Section 4.1, EmoryNLP: "iis" -> "is", MELD: "enrichadding to the richness""*
>
> Thank you for your correction.
>
> #### **Re** *Typos 5:*
>
> *"Table 2: It is confusing that the third column for Ubuntu doesn't match the other two (instead ED it reports AR), please highlight this in the caption or the header. Also please spell out what SOTA refers to, e.g. in the caption."*
>
> It depends on the dataset. EmoryNLP and MELD datasets provide emotion labels and can therefore be used to do ED tasks. The opposite Ubuntu IRC dataset provides Addressee tags and can therefore be used to do AR tasks.
>
> #### **Re** *Typos 6:*
>
> *"I would find it easier if the settings had some kind of special syntax..."*
>
> Thank you for your advice. We will modify them in the revision.
>
> #### **Re** *Typos 7:*
>
> *"In the related work section you mention a number of LLMs but I did not understand why you picked only GPT-4 and ChatGPT to test."*
>
> 1. ChatGPT and GPT-4 are recognized representatives of LLMs, so we only consider them to evaluate solving MPC problems.
> 2. For a reference, we provide the performance of Vicuna-1.5 (*w/. Speaker*) which is finetuned from Llama-2 on the MELD dataset as below. We found that it's much more difficult to control the output for Vicuna comparing with ChatGPT.
>
> |                                | MELD    |          |            |             |
> | ------------------------------ | ------- | -------- | ---------- | ----------- |
> | Vicuna-1.5-13b (*w/. Speaker*) | ED (F1) | SI (ACC) | RS (R10@1) | RG (METEOR) |
> | **Zero-Shot**                  | 54.89   | 42.79    | 11.29      | 12.43       |
> | **2-shots (Retrival)**         | 56.32   | 33.48    | 15.63      | 13.33       |
> | **2-shots (Random)**           | 55.26   | 25.58    | 25.08      | 11.82       |
>
> #### **Re** *Typos 8:*
>
> *"I found "addressee recognition" to be confusing... I was initially confused about the first utterance not having an addressee ..."*
>
> 1. This expression is from MPC-BERT [4].
> 2. In the Ubuntu scenario, since the first person posting doesn't target a specific person, it can be considered to not have an addressee.
>
> #### **References**
>
> [1] Is ChatGPT Good at Search? Investigating Large Language Models as Re-Ranking Agent. Sun et al. Arxiv 2023.
>
> [2] Is ChatGPT a General-Purpose Natural Language Processing Task Solver? Qin et al. Arxiv 2023.
>
> [3] MELD: A multimodal multi-party dataset for emotion recognition in conversations. Poria et al. ACL 2019.
>
> [4] MPC-BERT: A Pre-Trained Language Model for Multi-Party Conversation Understanding. Gu et al. ACL 2021.
>
> [5] GSN: A Graph-Structured Network for Multi-Party Dialogues. Hu et al. IJCAI 2019.
>
> [6] HeterMPC: A Heterogeneous Graph Neural Network for Response Generation in Multi-Party Conversations. Gu et al. ACL 2022

---

### Official Review · Reviewer_pXrH · 2023-08-06

**Soundness:** 4

**Excitement:**

2: Mediocre: This paper makes marginal contributions (vs non-contemporaneous work), so I would rather not see it in the conference.

**Paper Topic And Main Contributions:**

The paper presents experimental results with the use of ChatGPT / GPT-4 on multi-party conversation data. More precisely, it investigates the zero-shot performance of GPT3.5 and GPT-4 on various MPC tasks such as emotion detection, addresse recognition, speaker identification, response selection and generation, using three distinct datasets. The paper also shows that incorporating speaker and addressee information to the formatting of the dialogue history leads to performance improvements.

**Questions For The Authors:**

- is there a reason for not providing results with few-shots learning? This would be reasonably easy to test, by including e.g. a few examples into the prompt.

**Reasons To Accept:**

- the paper is well-written
- the problem of understanding multi-party conversation is both interesting and useful
- the experimental design is generally well described

**Reasons To Reject:**

- the contribution is very limited: it essentially boils down to providing zero-shot results with GPT 3.5 and 4 on existing tasks and datasets. The presented approach simply adds speaker and/or addressee into the formatting of the dialogue, but this is relatively trivial.
- the experiments only test for GPT 3.5 and GPT4 along with a few baselines based on BERT, BART and a few others. Why not include a few other large LLMs, such as Llama 2?

**Reproducibility:**

4: Could mostly reproduce the results, but there may be some variation because of sample variance or minor variations in their interpretation of the protocol or method.

**Reviewer Confidence:**

4: Quite sure. I tried to check the important points carefully. It's unlikely, though conceivable, that I missed something that should affect my ratings.

**Typos Grammar Style And Presentation Improvements:**

- line 22: treatise --> study
- line 29: graphical information flows?
- figure 1 can be dropped
- I would suggest another than than "MPC structure incorporation" for the approach
- lines 254-256:  I assume the last part (with "[utterance]" ...) should be removed?

---

> ### Author Rebuttal · Authors · 2023-08-29
>
> We appreciate your feedback and insights.
>
> #### **Re** *Reason To Reject 1*:
>
> *"the contribution is very limited: it essentially boils down to providing zero-shot results with GPT 3.5 and 4 on existing tasks and datasets. The presented approach simply adds speaker and/or addressee into the formatting of the dialogue, but this is relatively trivial."*
>
> 1. Since ChatGPT is a closed source model, we only have access to its inputs as well as outputs. So what we can do is change its inputs (Prompts) to get better results.
> 2. The main purpose of this paper is to explore whether the most powerful language models available can be good tools for solving MPC problems, which have not yet been explored. As the reviewer 2d2w agrees, the application of LLMs in MPC is expected to expand in the near future, and we believe that this paper can provide good inspiration for follow-up work.
> 3. MPCs do not maintain sequential information flows since the third utterance can reply to the first one, which brings a challenge for autoregressive serialized LLMs, especially for ChatGPT, which is carefully designed for two-person conversations.
> 4. As for how to improve the performance of LLMs in MPC, inspired by MPC--BERT [1], we utilize additional information including the speaker and the addressee. How these two types of information are utilized and how they affect the performance of LLMs in MPC tasks are also our contributions. In addition, we discuss the experimental results in detail. This discussion forms the basis for continued investigations into improving LLM effectiveness in MPC contexts.
>
> #### **Re** *Reason To Reject 2*:
>
> *"the experiments only test for GPT 3.5 and GPT4 along with a few baselines based on BERT, BART and a few others. Why not include a few other large LLMs, such as Llama 2?"*
>
> 1. ChatGPT and GPT-4 are well-recognized representatives of LLMs, so we only consider them as evaluation objects.
> 2. We wish to explore whether LLMs with zero-shot can perform as well as supervised trained models in the MPC task, so we do not consider other LLMs as evaluation objects.
> 3. For a reference, we provide the performance of Vicuna-1.5 (*w/. Speaker*) which is finetuned from Llama-2 on the MELD dataset as below. We found that it's much more difficult to control the output for Vicuna comparing with ChatGPT.
>
> |                                | MELD    |          |            |             |
> | ------------------------------ | ------- | -------- | ---------- | ----------- |
> | Vicuna-1.5-13b (*w/. Speaker*) | ED (F1) | SI (ACC) | RS (R10@1) | RG (METEOR) |
> | **Zero-Shot**                  | 54.89   | 42.79    | 11.29      | 12.43       |
> | **2-shots (Retrival)**         | 56.32   | 33.48    | 15.63      | 13.33       |
> | **2-shots (Random)**           | 55.26   | 25.58    | 25.08      | 11.82       |
>
>
>
> #### **Re** *Question:*
>
> *is there a reason for not providing results with few-shots learning? This would be reasonably easy to test, by including e.g. a few examples into the prompt.*
>
> 1. The scenario we consider is an out-of-the-box MPC solver, so we focus more on the performance of LLMs with zero-shot.
> 2. Few-shots increase the prompt tokens leading to more cost.
> 3. For comparison purposes, we also provide some results of the few-shots experiments as below. Example selection includes random-select and retrieval by similarity (calculated by Model sentence-transformers/all-MiniLM-L6-v2).
>    Note that *w/. All* means that *w/. Speaker & Addressee* if *Addressee* is accessable else *w/. Speaker*.
>
> |                       | MELD    |          |            |             | Ubuntu   IRC |          |            |             |
> | --------------------- | ------- | -------- | ---------- | ----------- | ------------ | -------- | ---------- | ----------- |
> |                       | ED (F1) | SI (ACC) | RS (R10@1) | RG (METEOR) | ED (F1)      | SI (ACC) | RS (R10@1) | RG (METEOR) |
> | **Zero-Shot**         |         |          |            |             |              |          |            |             |
> | GPT-3.5-Turbo w/. All | 60.90   | 57.67    | 39.17      | 9.86        | 67.19        | 38.50    | 29.50      | 12.07       |
> | GPT-4 w/. All         | 64.18   | 78.60    | 57.46      | 13.06       | 82.50        | 58.00    | 72.00      | 13.30       |
> | **2-shot (Retrival)** |         |          |            |             |              |          |            |             |
> | GPT-3.5-Turbo w/. All | 59.98   | 60.00    | 40.00      | 11.09       | 65.37        | 47.50    | 40.00      | 17.42       |
> | GPT-4 w/. All         | 64.84   | 80.00    | 54.00      | 13.42       | 81.00        | 86.00    | 78.00      | 20.40       |
> | **2-shot (Random)**   |         |          |            |             |              |          |            |             |
> | GPT-3.5-Turbo w/. All | 61.88   | 61.39    | 38.21      | 8.16        | 65.35        | 41.50    | 24.00      | 12.26       |
> | GPT-4 w/. All         | 63.65   | 86.00    | 54.00      | 13.24       | 79.76        | 68.00    | 70.00      | 9.99        |
>
>
>
> #### **Re** *Typos 1:*
>
> *"line 22: treatise --> study"*
>
> Thank you for your correction.
>
> #### **Re** *Typos 2:*
>
> *"line 29: graphical information flows? "*
>
> MPCs do not maintain sequential information flows since the third utterance can reply to the first one. The same term also was used in HeterMPC[2].
>
> #### **Re** *Typos 3:*
>
> *"figure 1 can be dropped"*
>
> Thank you for your advice. Figure 1 presents a vivid example of human-machine dialogue that demonstrates the power of current LLMs. In terms of content, ChatGPT's response was intriguing and touched upon some of the critical aspects we are investigating.
>
> We will remove figure 1 and put the example in the main body in our revision.
>
> #### **Re** *Typos 4:*
>
> *"I would suggest another than than "MPC structure incorporation" for the approach"*
>
> Thank you for your advice. As stated in *Re Typos 2*, MPC is with graph information flow. "Speaker to Utterance" and "Utterance to addressee" can both be considered as “graph structure".
>
> #### **Re** *Typos 5:*
>
> *"lines 254-256: I assume the last part (with "[utterance]" ...) should be removed?"*
>
> I apologize for the lack of clarity. The meaning of this sentence is to add speaker information to utterances. We will correct the expression of (with "[utterance]" ...) to (and "[utterance]" …).
>
> #### **References**
>
> [1] MPC-BERT: A Pre-Trained Language Model for Multi-Party Conversation Understanding. Gu et al. ACL 2021.
>
> [2] HeterMPC: A Heterogeneous Graph Neural Network for Response Generation in Multi-Party Conversations. Gu et al. ACL 2022

---

### Meta-Review · Area_Chair_ccN6 · 2023-09-23

**Recommendation:** 3

**Metareview:**

*Summary*: This work evaluates the zero-shot capabilities of ChatGPT and GPT-4 on multi-party conversation (MPC) tasks. Five tasks including emotion detection, accurately detecting the addresses in a conversation, speaker identification, response selection and generation have been explored. They find that including speaker and addressee information in the prompt yields performance improvements.

*Evaluation*:  R1 and R3 rated this work as medium to strong on soundness (3/4), while excitement ratings were lower (2/3). R2 rated this low on soundness (2) and stated two main concerns: A) Since training data of ChatGPT, GPT-4 is unknown, we cannot be sure there is no test leakage. B) No in-depth analysis of failure cases has been presented. Regarding A), this is a broader concern that holds for all closed model evaluations and not specific to this work. B) is indeed a valid concern. Given this is a first investigation of the capabilities of ChatGPT/GPT-4 on MPC tasks, demanding an in-depth analysis of the failure cases (along with anecdotal examples in an Appendix) is warranted and would have been useful for the reader.

---

### Decision · Program_Chairs · 2023-10-07

**Decision:**

Accept-Findings

**Comment:**

*Summary*: This work evaluates the zero-shot capabilities of ChatGPT and GPT-4 on multi-party conversation (MPC) tasks. Five tasks including emotion detection, accurately detecting the addresses in a conversation, speaker identification, response selection and generation have been explored. They find that including speaker and addressee information in the prompt yields performance improvements.

*Evaluation*:  R1 and R3 rated this work as medium to strong on soundness (3/4), while excitement ratings were lower (2/3). R2 rated this low on soundness (2) and stated two main concerns: A) Since training data of ChatGPT, GPT-4 is unknown, we cannot be sure there is no test leakage. B) No in-depth analysis of failure cases has been presented. Regarding A), this is a broader concern that holds for all closed model evaluations and not specific to this work. B) is indeed a valid concern. Given this is a first investigation of the capabilities of ChatGPT/GPT-4 on MPC tasks, demanding an in-depth analysis of the failure cases (along with anecdotal examples in an Appendix) is warranted and would have been useful for the reader.